# Ventrointermediate thalamic stimulation improves motor learning in humans
Angela Voegtle [1] ✉, Laila Terzic[1], Amr Farahat[1,2], Nanna Hartong[3], Imke Galazky[3], Hermann Hinrichs[3,4,5], Slawomir J. Nasuto[6], Adriano de Oliveira Andrade [7], Robert T. Knight [8], Richard B. Ivry[8,9], Jürgen Voges[10], Matthias Deliano[11], Lars Buentjen [10] & Catherine M. Sweeney-Reed [1,5] ✉

Ventrointermediate thalamic stimulation (VIM-DBS) modulates oscillatory activity in a cortical network including primary motor cortex, premotor cortex, and parietal cortex. Here we show that, beyond the beneficial effects of VIM-DBS on motor execution, this form of invasive stimulation facilitates production of sequential finger movements that follow a repeated sequence. These results highlight the role of thalamo–cortical activity in motor learning.

Motor learning has been associated with a network that includes the cerebellum, primary motor cortex (M1), basal ganglia, ventrointermediate nucleus of the thalamus (VIM), and parietal–frontal cortex[1–5]. This network is disrupted in patients with essential tremor (ET)[6,7], and deep brain stimulation of the VIM (VIM-DBS) has been shown not only to alleviate tremor, but also can improve motor learning[4,8]. To date, the neural mechanisms associated with this improvement are not well understood. Here we investigate the impact of VIM-DBS on oscillatory activity in cortical nodes of the motor learning network during sequence learning using the well-established serial reaction time task (SRTT)[5,9–12]. Modulation of oscillatory brain activity has been observed in the alpha (8–12 Hz)[5,10,11] and beta bands (13–30 Hz)[12] during motor sequence learning. For example, patients with Parkinson's disease show enhanced and prolonged beta band power decrease, compared to controls. This enhanced suppression correlated with impairments in motor sequence learning[12]. ET patients also exhibit enhanced suppression in the alpha/beta band during movement, comparable to Parkinson's disease patients[6]. We hypothesized that oscillatory power is modulated by VIM-DBS in patients with ET, including a reduction of pathological alpha/beta band power suppression. Note that a reduction in alpha/beta suppression is distinct from an increase in power. Given that sequence learning is modulated by VIM-DBS[4], we predicted that a reduction in alpha/beta band power suppression would accompany improved performance of a repeated sequence of finger movements.

We examined a behavioral index of motor sequence learning and scalp electroencephalography (EEG) oscillatory activity during SRTT performance, comparing conditions in which the VIM-DBS was ON or OFF. The task required the participants to make a series of finger responses, which either followed a repeated sequence or were random. As our measure of learning, we derived a normalized learning score for each block, contrasting reaction times (RTs) for repeated and random sequences within the block. We then examined oscillatory activity during the SRTT, to investigate whether VIM-DBS has an impact on established oscillatory correlates of motor sequence learning. We focused on stimulus-locked activity, using artifact removal methods[13] that enabled analysis of EEG data recorded during DBS. We used cluster-based permutation tests[14] to examine oscillatory spectral power from 59 electrodes over an epoch starting 200 ms before to 1200 ms after stimulus onset, with the analysis spanning a frequency range of 2–30 Hz. Based on the behavioral results, we examined the general effects of DBS across all trials (repeated and random) at the end of training (Block 4). We then performed separate analyses for repeated and random trials, contrasting the ON-OFF conditions at the end of training. To account for the effects of time, we contrasted trials from Block 4 to Block 1 separately for the repeated and the random sequences, and compared the DBS-ON against DBS-OFF conditions.

Based on previous findings of a greater impact on motor sequence learning when VIM electrodes were located more laterally[4], we assessed whether any modulation of oscillatory power by VIM-DBS during motor sequence learning was dependent on the specific electrode location within

[1]Neurocybernetics and Rehabilitation, Department of Neurology, Otto von Guericke University Magdeburg, Magdeburg, Germany. [2]Ernst Strüngmann Institute for Neuroscience in Cooperation with Max Planck Society, Frankfurt am Main, Germany. [3]Department of Neurology, Otto von Guericke University Magdeburg, Magdeburg, Germany. [4]Department of Behavioral Neurology, Leibniz Institute for Neurobiology, Magdeburg, Germany. [5]Center for Behavioral Brain Sciences—CBBS, Otto von Guericke University Magdeburg, Magdeburg, Germany. [6]Biomedical Sciences and Biomedical Engineering Division, School of Biological Sciences, University of Reading, Reading, UK. [7]Faculty of Electrical Engineering, Center for Innovation and Technology Assessment in Health, Postgraduate Program in Electrical and Biomedical Engineering, Federal University of Uberlândia, Uberlândia, Brazil. [8]Helen Wills Neuroscience Institute, University of California—Berkeley, Berkeley, CA, USA. [9]Department biof Psychology, University of California—Berkeley, Berkeley, CA, USA. [10]Department of Stereotactic Neurosurgery, Otto von Guericke University Magdeburg, Magdeburg, Germany. [11]Combinatorial Neuroimaging Core Facility, Leibniz Institute for Neurobiology, Magdeburg, Germany. ✉e-mail: angela.voegtle@med.ovgu.de; catherine.sweeney-reed@med.ovgu.de

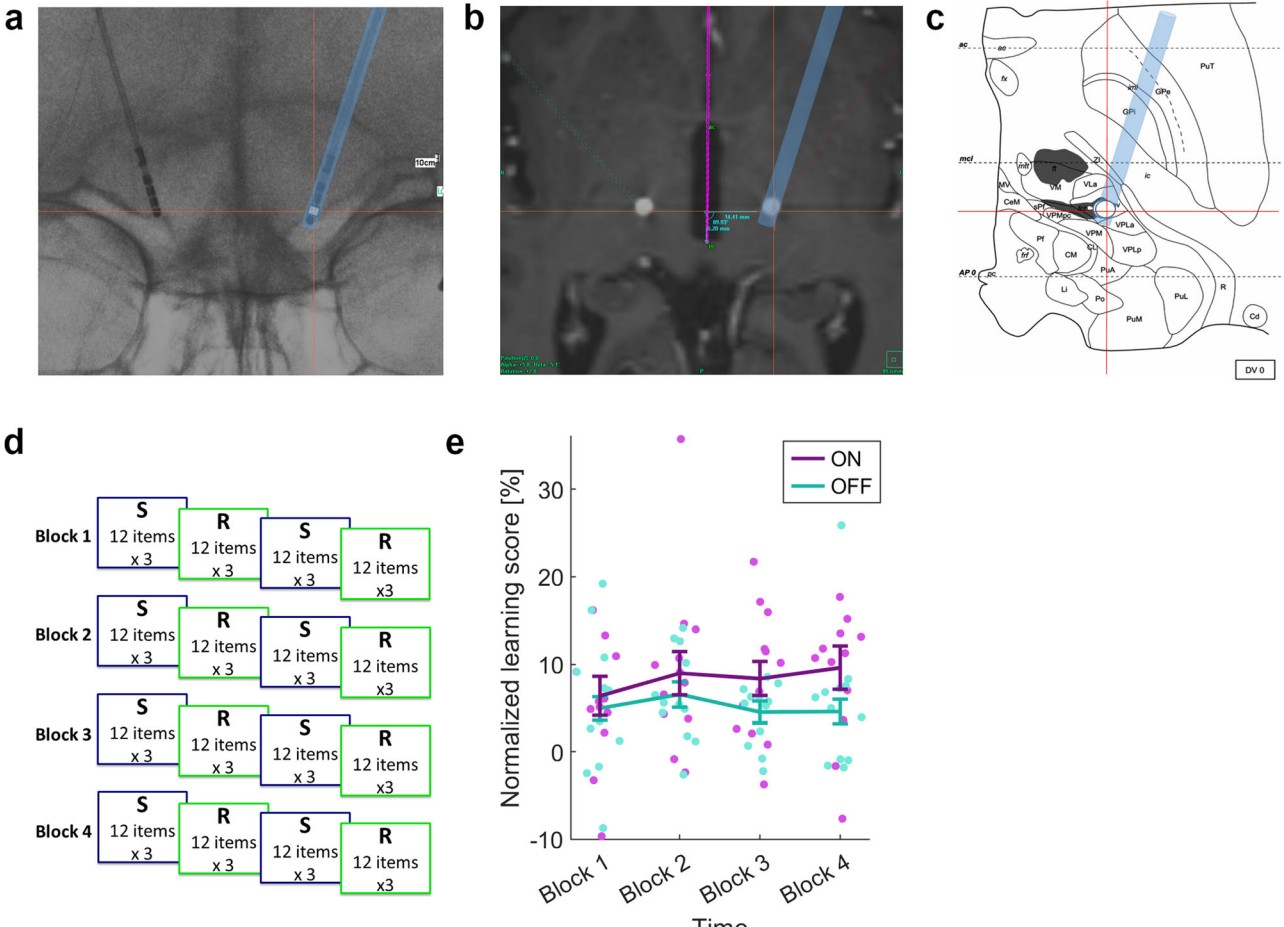

**Fig. 1 | Illustration of stereotactic targeting of ventrointermediate nucleus of the thalamus (VIM), experimental design, and impact of deep brain stimulation of the VIM on motor sequence learning. a** Stereotactic x-ray depicting intraoperative electrode location. Red crosshair: contact in the VIM (Vlpv-Morel[29]).
**b** Postoperative CT co-registered to preoperative magnetic resonance imaging to establish electrode placement. White circle: CT-artifact of electrode contact in the VIM. Purple line: trajectory through the intracommissural line (IL). Red crosshair: electrode position in co-registered intraoperative x-ray. **c** Stereotactic atlas of the thalamus[30] (line drawing from ref. 30: Copyright © 2007. From 'Stereotactic Atlas of the Human Thalamus and Basal Ganglia' by A. Morel. Reproduced by permission of Taylor and Francis Group, LLC, a division of Informa plc) at IL-level. Red crosshair: represents electrode position in relation to the IL. **d** Experiment: serial reaction time task, with alternating runs of repeated sequences (S) and random (R) trials, in 4 blocks. **e** Interaction for the normalized learning score between *Stimulation Mode* (DBS-ON, DBS-OFF) and *Time* (Block 1 to Block 4), showing estimated marginal means and standard errors over individual scores (*n* = 12).

the VIM. In addition, we evaluated whether power modulation was related to the degree of tremor amelioration or the total energy delivered through DBS.

## Results
The data from 12 participants were included in all analyses. For the normalized learning score, a three-way interaction was observed between *Stimulation Mode* (DBS-ON, DBS-OFF), *Time* (Block 1 to Block 4), and *Tremor Score Difference* ($F(3,24) = 5.84$, $p = 0.004$; $\eta^2 = 0.42$), and correcting for *Tremor Score Difference*, the two-way interaction between *Stimulation Mode* and *Time* remained significant ($F(3,24) = 3.70$, p = 0.025, $\eta^2 = 0.32$; Fig. 1). The normalized learning score was greater for DBS-ON compared to DBS-OFF in all four blocks. Pairwise post hoc testing showed that the only significant difference between DBS-ON and DBS-OFF was a greater learning score when DBS was on in Block 4 (DBS-ON: $M = 9.59$, 95% CI [3.89 15.3]; DBS-OFF: $M = 4.61$, 95% CI [1.33 7.89]; $p = 0.035$). A two-way interaction was also found between *Tremor Score Difference* and *Time*, with patients with the greatest improvement in tremor score showing the greatest improvement in learning score over time ($F(3,24) = 3.38$, $p = 0.035$, $\eta^2 = 0.30$) but not between *Tremor Score Difference* and *Stimulation Mode*. No significant three-way interactions were found between *Stimulation Mode*, *Time*, and *Stimulation Order*, or between *Stimulation Mode*, *Time*,

and *Total Electrical Energy Delivered*. Baseline RTs to the random sequence in Block 1 did not differ according to whether DBS was on ($M = 633.5$, SD 111.3) or off ($M = 650.3$, SD 111.6; $T(11) = -0.89$, $p = 0.40$). All patients experienced improvement in tremor score with DBS-ON compared with DBS-OFF.

Turning to the physiological data, when we collapsed across the repeated sequence and random trials at the end of training (Block 4), we observed a significant difference between DBS-ON and DBS-OFF. The cluster was observed over bilateral sensorimotor areas (cluster-t = 9712, $p_{pos} = 0.004$, SD = 0.003, Cohen's $d = 2.156$), spanning a time window of ~500 ms, starting ~500 ms after stimulus onset, in the alpha/beta range (~6–16 Hz). The cluster indicated a widespread, bilateral reduced suppression in alpha/beta power when VIM-DBS was on compared with off (Fig. 2).

At the end of training, a difference between DBS-ON and DBS-OFF was observed for the repeated sequence, with a cluster encompassing the alpha/beta frequency bands. These effects were found over the central and ipsilateral motor area, including M1 and premotor cortex (PMC; Fig. 3), spanning a time window of ~320 ms, starting ~580 ms after stimulus onset (cluster-t = 5660, $p_{pos} = 0.008$, SD = 0.004, Cohen's $d = 1.779$). During DBS-ON, alpha/beta power was less suppressed than during DBS-OFF (Fig. 3). No difference was detected between DBS-ON and DBS-OFF during

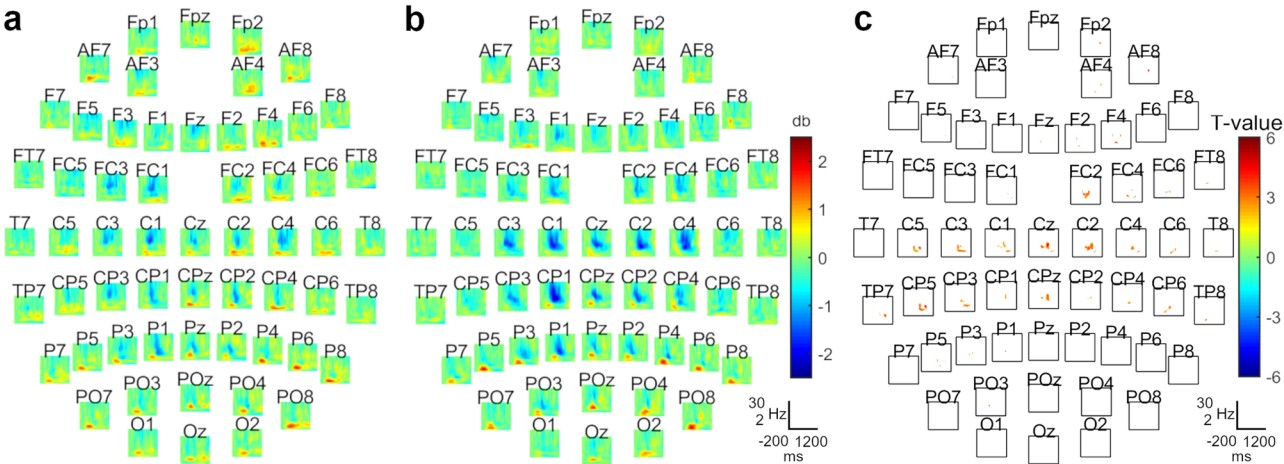

**Fig. 2 | General impact of stimulation at the end of training (n = 12). a** Grand-average during DBS-ON. **b** Grand-average during DBS-OFF. **c** Location of the observed cluster, when testing DBS-ON against DBS-OFF over all trials (cluster-$t = 9712$, $p_{pos} = 0.004$, SD = 0.003, Cohen's $d = 2.156$). The cluster included electrodes spanning the bilateral sensorimotor areas. **a**, **b**, **c** x-axis = Time: −200 to 1200 ms; y-axis = Frequency: 2–30 Hz.

**Fig. 3 | Oscillatory power during responses to repeated and random motor sequences at the end of training (n = 12). a** Grand-average difference between DBS-ON and DBS-OFF for repeated sequences. **b** Grand-average difference between DBS-ON and DBS-OFF for random trials. **c** Power values within the determined cluster were averaged over time, frequency, and electrodes for both stimulation modes and sequence types. Interaction for power values between *Stimulation Mode* (DBS-ON, DBS-OFF) and *Sequence Type* (Repeated, Random), showing means and standard errors. **d** Location of the observed cluster, when testing DBS-ON against DBS-OFF during repeated sequences (cluster-$t = 5660$, $p_{pos} = 0.008$, SD = 0.004, Cohen's $d = 1.779$). The cluster included electrodes spanning the ipsilateral sensorimotor cortex. The channel contributing most to the cluster was C2. **e** Left panel: grand-average of channel C2 during DBS-ON; right panel: grand-average of channel C2 during DBS-OFF. **a**, **b**, **d** x-axis = Time: −200 to 1200 ms; y-axis = Frequency: 2–30 Hz.

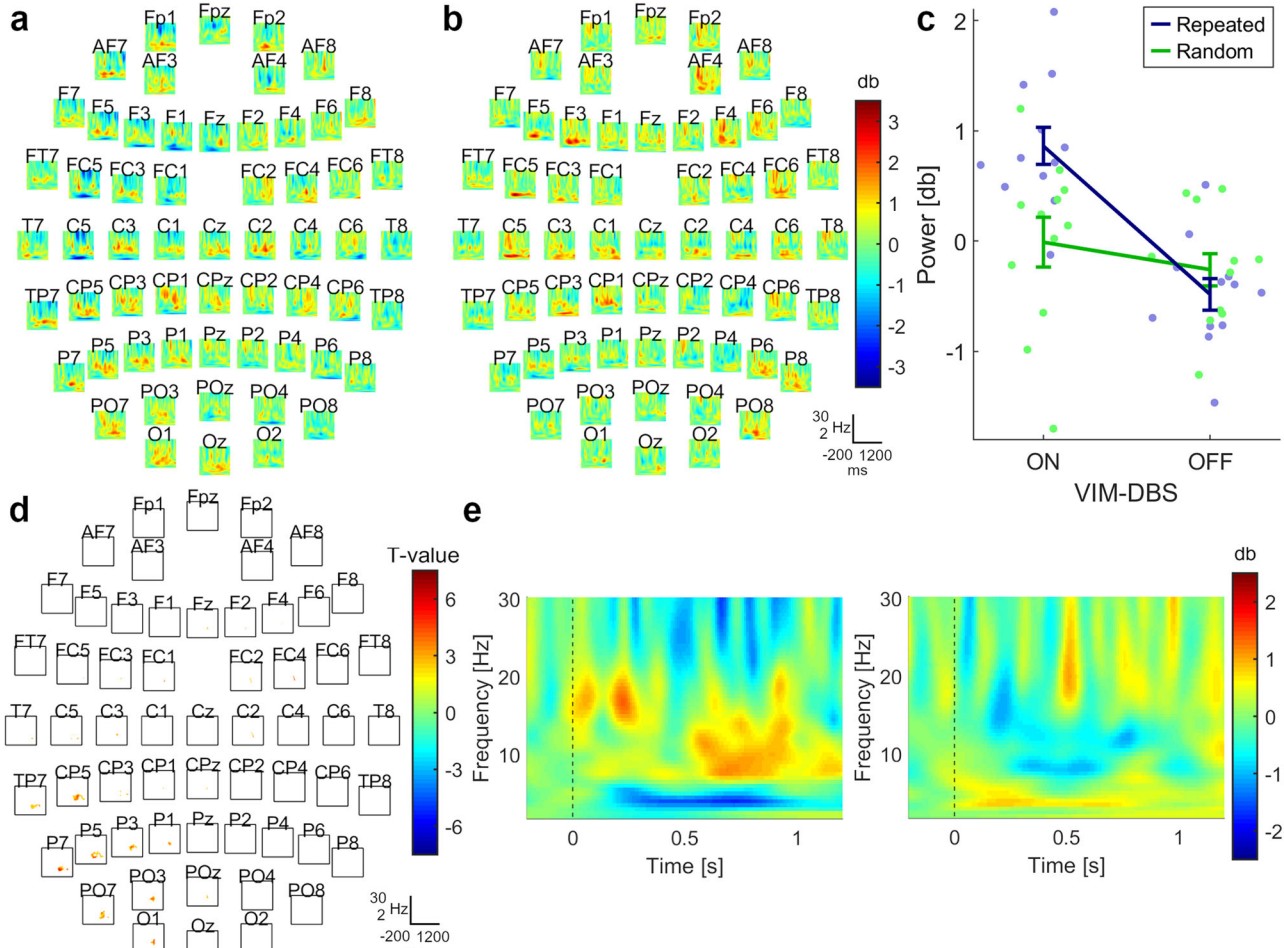

**Fig. 4 | Changes in oscillatory power over the course of time (difference between Block 4 and Block 1; n = 12).** **a** Grand-average difference between DBS-ON and DBS-OFF for the contrast between Block 4 and Block 1 during repeated sequences. **b** Grand-average difference between DBS-ON and DBS-OFF for the contrast between Block 4 and Block 1 during random trials. **c** Power values within the determined cluster were averaged over time, frequency, and electrodes for both stimulation modes and sequence types. Interaction for power values between *Stimulation Mode* (DBS-ON, DBS-OFF) and *Sequence Type* (Repeated, Random), showing means and standard errors. **d** Location of the observed cluster, representing significant difference between DBS-ON and DBS-OFF for the contrast over time during repeated sequences (cluster-t = 7237, $p_{pos}$ = 0.012, SD = 0.005, Cohen's *d* = 2.485). The cluster included electrodes spanning the contralateral cortex over a central-parietal, parietal-occipital region. The channel contributing most to the cluster was CP5. **e** Left panel: grand-average of channel CP5 during DBS-ON; right panel: grand-average of channel CP5 during DBS-OFF. **a, b, d** x-axis = Time: −200 to 1200 ms; y-axis = Frequency: 2–30 Hz.

random trials (cluster-t = 1407, $p_{pos}$ = 0.325, SD = 0.021; cluster-t = −88, $p_{neg}$ = 0.994, SD = 0.004). A repeated measures ANOVA showed an interaction between *Stimulation Mode* (DBS-ON, DBS-OFF) and *Sequence Type* (Repeated, Random) ($F(1,11)$ = 9.84, $p$ = 0.009, $\eta^2$ = 0.47). Post hoc tests showed that the power difference during repeated sequences compared to random sequences was greater with DBS-ON (Repeated: *M* = 0.22, 95% CI [−0.18, 0.62]; Random: *M* = −0.38, 95% CI [−0.74, −0.02]) compared to DBS-OFF (Repeated: *M* = −0.95, 95% CI [−1.39, −0.51]; Random: *M* = −0.69, 95% CI [−1.19, −0.20]) ($T(11)$ = 3.14, $p$ = 0.009, Cohen's *d* = 0.91) (Fig. 3).

We also observed a difference between DBS-ON and DBS-OFF for the contrast between Block 4 and Block 1 during repeated sequences (cluster-t = 7237, $p_{pos}$ = 0.012, SD = 0.005, Cohen's *d* = 2.485). The cluster included the alpha frequency band and was present over a central–parietal–occipital area contralateral to movement, starting ~500 ms after stimulus onset and persisting until ~800 ms after stimulus onset; at some channel locations, up to ~1100 ms after stimulus onset (Fig. 4). Again, no difference was detected between the ON and OFF conditions during random trials (cluster-t = 2284, $p_{pos}$ = 0.140, SD = 0.016; cluster-t = −241, $p_{neg}$ = 0.884, SD = 0.014). A repeated measures ANOVA showed an interaction between *Stimulation Mode* (DBS-ON, DBS-OFF) and *Sequence Type* (Repeated, Random;

$F(1,11)$ = 10.06, $p$ = 0.009, $\eta^2$ = 0.48). Post hoc tests showed that the increase in power during repeated sequences compared to random sequences was greater with DBS-ON (Repeated: *M* = 0.86, 95% CI [0.49, 1.23]; Random: *M* = −0.011, 95% CI [−0.51, 0.49]) compared to DBS-OFF (Repeated: *M* = −0.48, 95% CI [−0.80, −0.171]; Random: *M* = −0.26, 95% CI [−0.59, 0.066]) ($T(11)$ = 3.17, $p$ = 0.009, Cohen's *d* = 0.92) (Fig. 4).

Multiple linear regression was used to test whether the location of VIM-DBS stimulation predicted mean power over time, frequency, and EEG channel within the cluster, taking into account stimulation intensity and tremor severity. The overall regression was significant ($R^2$ = 0.886, $F(5,65)$ = 9.319, $p$ = 0.009). Mean power was predicted by the AC–PC-x and AC–PC-y coordinates of the VIM stimulation contact, the total electrical energy delivered to the VIM, and tremor score, but not by the AC–PC-z coordinate (Supplementary Table 1). The power was greater when the VIM electrode was more lateral and anterior (Supplementary Fig. 1).

## Discussion
Motor sequence learning improved when VIM-DBS was on compared to off, with the learning boost during stimulation becoming significant in the final block. RTs in response to the random stimuli did not differ according to whether DBS was on or off, suggesting that DBS-ON enhanced motor

sequence learning rather than resulted in a general improvement in motor execution. The absence of an interaction between *Tremor Score Difference* and *Stimulation Mode* suggests that change in tremor score does not fully account for the motor learning score improvement.

Over the course of time when DBS was on, compared with off, we observed an increase in alpha power in a cluster centered over left parietal cortex, contralateral to the moving hand. Note that, in contrast to motor cortical regions, movement-related suppression of an ongoing alpha/beta rhythm has not been reported in the parietal cortex, and we therefore refer to a power increase. The contralateral parietal cortex contributes to motor sequence learning[15] by integrating visual and somatosensory inputs[1,16]. Visuomotor control is established together with outputs to the dorsal PMC[16], resulting in output to M1 for movement generation[1]. ET patients show decreased bidirectional functional connectivity between the parietal and contralateral motor cortex[7]. The increase in contralateral parietal alpha power observed when the stimulation was on may facilitate information integration between premotor and parietal areas as a sequence is learned through repetition.

The reduced alpha suppression we observed at the end of training of a repeated sequence over central and ipsilateral motor cortical regions during VIM-DBS is consistent with the re-establishment of motor learning processes that were disrupted by pathological activity in this network. When examining the data pooled across repeated and random trials, VIM-DBS was associated with a widespread, bilateral reduced suppression in alpha/beta power, compared to when the stimulation was off. This general effect of VIM-DBS on cortical oscillatory activity appears to have a specific impact on motor sequence learning performance, given the effect on the normalized learning score. The effect on oscillatory power was greater when the stimulation site was at more lateral aspects of the VIM. Moreover, we previously observed enhanced motor sequence learning performance with VIM-DBS delivery to more lateral compared to medial VIM sites[4]. There are cytoarchitectural and connectivity differences between lateral and medial aspects of VIM[17,18]. Physiologically, single-cell recordings from the lateral but not medial VIM show tactile and kinesthetic responses[19,20], which is relevant for motor learning. Of these recordings, however, 68% of the kinesthetic neurons related to the upper limb and 12% to the more medially represented face location[20]. The somatotopic organization of the VIM, with the upper limb represented more laterally than the face[21,22], could account for their finding of responses only in lateral VIM and our finding that the lateral electrode signals were associated with motor sequence learning tested in the upper limb to a greater extent than medial electrode signals. Previous studies have shown that the alpha suppression over M1 typically observed during movement is attenuated when a movement sequence is repeated. This pattern has been hypothesized to reflect reduced attentional processing once the sequence was established[10,11]. On the other hand, a local increase in alpha band power has also been thought to have a time-dependent inhibitory effect on cortical functioning, specifically in regions no longer involved in task performance[23]. Future work will be needed to establish whether the reduction in alpha suppression during VIM-DBS indicates lower attentional requirements after a sequence is learned, or whether it reflects an enhancement of selective attention, with greater inhibition of task-irrelevant brain regions.

Beta band modulations may reflect cortical reorganization associated with sequence learning[10]. While beta suppression has been associated with diminishing interference effects by stabilizing the newly learned sequence during early consolidation[10], prolonged beta band activity suppression is thought to hinder behavioral flexibility by promoting the maintenance of the current motor state, even if this is not optimal[24]. Greater beta suppression has been found in patients with Parkinson's disease at the end of training with diminished learning[12]. In our ET patient group, DBS-ON led to lower beta band suppression at the end of training over the ipsilateral motor cortex and accompanied improved learning.

Interestingly, while the reduction in alpha/beta sensorimotor suppression over motor cortex during stimulation was bilateral for all trials, the effect for repeated sequences was unilateral and ipsilateral to movement.

Given that testing was limited to the right hand, we cannot say if this asymmetry (or the one noted in the parietal cortex) is related to the relationship of the cortex and moving hand (e.g., ipsilateral vs contralateral) or hemispheric specialization. It does suggest that the observed impact was on higher level processes associated with motor sequence learning rather than motor execution per se. In terms of hemispheric differences, previous work has shown the engagement of right PMC during spatial tasks as well as in the later stages of sequence learning[3].

We have suggested that VIM-DBS produces a change in oscillatory power and that these changes contribute to the observed improvement in learning. We recognize that there are other, less direct ways in which these changes might come about from VIM-DBS. For example, with the tremor reduction observed during VIM-DBS, the patients might have more attentional processes to devote to extracting the stimulus pattern during the sequence blocks. On the other hand, although the regression indicated some relationship between oscillatory power and tremor score, the latter did not fully account for the difference in mean learning score with DBS-ON compared with DBS-OFF, suggesting that neither general motor improvement nor changes in attentional state fully explain the improved motor sequence learning. We note that while attention improvement could theoretically accompany tremor amelioration, we did not directly test attention here.

The present results provide direct evidence for the engagement of the VIM in motor sequence learning, showing modulation of electrophysiological activity in nodes of the motor learning network through VIM-DBS[4]. We postulate that VIM-DBS interrupted pathophysiological activity in cortical networks, akin to the normalization of certain oscillatory patterns in M1 observed in Parkinsons' disease patients who receive subthalamic nucleus DBS[25]. Subcortical, high frequency DBS appears to be able to override disease-related oscillatory patterns. Future work is required to establish whether DBS promotes re-establishment of normal oscillatory activity, suppresses abnormal activity, or imposes different activity patterns that enable improved function.

## Methods
### Participants
Sixteen right-handed patients with VIM electrodes previously implanted for DBS treatment of ET were recruited through the Stereotactic Neurosurgery Department, University Hospital, Magdeburg and were tested in an out-patient setting with EEG recorded while performing the experiment. Four datasets were excluded due to noisy EEG and lack of responses, resulting in the inclusion of $n = 12$ patients (Supplementary Table 2). Disease severity was quantified at the time of the study using the Fahn-Tolosa-Marin Tremor Rating Scale[26] with stimulation on and off.

VIM-DBS electrode locations were determined relative to the anterior and posterior commissures (AC–PC line), based on co-registering the post-operative computed tomography scans and the electrode coordinates from the intraoperative stereotactic x-rays, with the pre-operative structural magnetic resonance images (Fig. 1, Supplementary Table 2) recorded using a Siemens Verio scanner (Siemens, Erlangen, Germany) equipped with a 32-channel head coil. The images were produced using the Inomed (Emmendingen, Germany) software package.

All participants gave written, informed consent prior to inclusion in the study, which was approved by the Local Ethics Committee of the Medical Faculty of the Otto von Guericke University Magdeburg, and carried out in accordance with the Declaration of Helsinki. All ethical regulations relevant to human research participants were followed.

### Study design
Each participant performed two sessions of the SRTT on the same day, one session with (DBS-ON) and one without DBS (DBS-OFF), in a counter-balanced order. The task was presented using Presentation software (Neurobehavioral Systems, Berkeley, CA, USA).

Participants placed four fingers of their dominant (right) hand on four buttons of an ergonomically shaped response button pad. A row of four

squares was shown on a computer screen. When a square turned red, participants were asked to respond by pressing the corresponding button on the response pad, with compatible S-R [stimulus–response] mapping, as quickly and accurately as possible. The squares were highlighted red according to a 12-item sequence at locations 1-3-2-1-4-1-2-3-1-3-2-4, or at random. The random presentation order was constrained, such that the same square was not highlighted consecutively, and each location was presented at least once every 12 items. The stimuli were presented for 500 ms, and the inter-stimulus interval was fixed at 1200 ms, independent of the participants' response times. Both sessions consisted of 4 blocks, each containing 144 trials. The 144 trials alternated twice between three repetitions of the 12-item sequence (36 trials) and 36 random trials, always starting with the sequential trials (Fig. 1). We refer to Block 4 of the experiment as the end of training.

## DBS

Each DBS electrode probe had four contact locations at which stimulation could be applied. When the task was performed with the DBS on, stimulation was applied at the contact location, amplitude, frequency, and pulse width determined by the specialist DBS nurse to provide optimal tremor suppression while minimizing side-effects (Supplementary Table 2). This approach was taken for two reasons: the clinically-determined optimal parameters reflect the realistic impact on motor sequence learning in patients receiving VIM-DBS for tremor suppression, and from an ethical standpoint, only stimulation was applied that had been determined in routine clinical practice to provide maximal benefit to the patients. For statistical comparison, we estimated the total electrical energy delivered[27] for both hemispheres.

## EEG recording and preprocessing

During the SRTT, EEG was recorded using a 64-electrode EEG cap (Brain Products, Gilching, Germany), with electrode AFz set as the ground and electrode FCz as the reference. The sampling rate was 500 Hz. EEG data were recorded with Brain Vision Recorder (Brain Products, Gilching, Germany) software, and the following offline data processing was conducted with MATLAB R2018a (MathWorks, Natick, MA, USA) and the toolboxes DBSFILT (version 0.18b)[13] and Fieldtrip (version 20180826)[14].

Raw data files were band-pass filtered from 1 Hz to 100 Hz and notch-filtered between 49 and 51 Hz using DBSFILT[13]. DBSFILT was then used to apply a Hampel filter to all datasets to remove aliasing peaks in lower frequencies, which are common DBS artifacts. Aliased frequencies can be detected following application of a Hampel filter, enabling removal of only the noise component at each interference frequency[13]. Further data processing was performed using Fieldtrip[14]. The data were segmented into epochs from 1400 ms before stimulus onset until 2600 ms after stimulus onset. The 200 ms before stimulus onset were used for baseline correction. Each highlighting of a square in red was considered a stimulus. Analysis of data time-locked to the stimulus is a common approach[5,12] and offers the advantage that it resolves contamination by the activity related to the motor response, as well as tremor activity. The data were again band-pass filtered from 1 Hz to 30 Hz, using a padding length of 10 s. Bad channels were removed from the data based on visual inspection. Independent component analysis was performed to eliminate eye blinks and residual DBS artifacts, so that these trials could be retained. Afterwards, all epochs with values exceeding $\pm 100\,\mu V$ were excluded. Previously removed bad channels were then replaced using spherical spline interpolation, omitting electrodes FT9, FT10, TP9, TP10, and IO due to noise. Surface Laplacians were calculated to reduce the effects of volume conduction and to improve the spatial resolution of the EEG. All subsequent analyses were applied to the transformed data.

Time–frequency decomposition was performed through convolution with five-cycle complex Morlet wavelets for the frequencies ranging from 2 Hz to 30 Hz in increments of 0.5 Hz over the time window from −200 ms to 1200 ms, in increments of 10 ms. The time–frequency data were normalized to the baseline window 200 ms before until stimulus onset. Event-

related spectral perturbations (ERSPs) were determined separately for each channel, block (Block 1, Block 4), stimulation mode (DBS-ON, DBS-OFF), sequence type (repeated, random), and participant by averaging over the trials in which the target was correctly identified. To examine how cortical oscillations potentially change over the course of motor sequence learning, a contrast was formed by subtracting participants' individual ERSP in Block 1 from that in Block 4 separately for stimulation and sequence type. For visualization, grand average ERSPs were then calculated across participants, and difference plots were created by subtracting the grand average DBS-OFF from DBS-ON.

## Statistics and reproducibility

**Behavioral analysis.** Statistical analysis was performed using IBM SPSS Statistics 23 (IBM, Armonk, NY, USA). To quantify learning, we derived a normalized learning score $\frac{\text{mean RT to random} - \text{mean RT to repeated}}{\text{mean RT to random}}$. This measure was calculated for each individual for each of the four blocks within a session (DBS-ON or DBS-OFF). This provides a way of looking at learning across the four blocks that is independent of general changes in RT, which might occur due to changes in the strength of the stimulus-response mapping or fatigue. The normalized learning scores were analyzed with a repeated measures ANOVA with the within-subject factors *Stimulation Mode* (DBS-ON, DBS-OFF) and *Time* (Blocks 1 to 4), and *Stimulation Order* (ON First, OFF First) as a between-subject factor. We included *Tremor Score Difference* (DBS-OFF - DBS-ON) and the *Total Electrical Energy Delivered* as covariates. Post hoc tests are reported following Bonferroni correction for multiple comparisons. 95% confidence intervals (95% CI) are provided. To evaluate whether simple RTs are affected by VIM-DBS, irrespective of learning, we applied a paired T-test to the baseline RTs to the random sequence in Block 1 during DBS-ON compared with DBS-OFF, when no disruption by learning would be expected.

**Electrophysiological analysis.** To identify effects of DBS on oscillatory spectral power, irrespective of task performance, and during the repeated and random sequences, we performed non-parametric cluster-based permutation tests[14], including all 59 channels, time points (200 ms before to 1200 ms after stimulus onset), and frequencies (2–30 Hz). The first analysis was applied to all trials at the end of training, including repeated and random trials. The second analysis also focused on the end of training, where we assumed the sequence to be maximally learned, using all successfully repeated trials during Block 4 to compare between DBS-ON and DBS-OFF conditions. We performed an analogous analysis for random trials. To account for the effects of time on oscillatory power, we performed a third analysis, comparing the contrasted trials of Block 4 to Block 1 between DBS-ON and DBS-OFF conditions, for repeated trials. We performed an analogous analysis for random trials.

To confirm the specificity of the power differences between stimulation modes to sequence learning, the mean power values within the previously determined clusters were averaged over time, frequency, and electrodes during the repeated sequence, and analogous power values were calculated from the data recorded during presentation of stimuli in a random order. Repeated measures ANOVAs with factors *Stimulation Mode* (DBS-ON, DBS-OFF) and *Sequence Type* (Repeated, Random) were then applied, followed by post hoc tests.

Cluster-based permutation tests enabled analysis of the neuronal data without a priori assumptions regarding the exact location or extent of a possible effect. The multiple comparison problem is resolved by applying a single test statistic to clusters (adjacent points in space, time, and frequency, which differ between conditions at a pre-defined threshold) instead of evaluating the differences between conditions at each sample point separately. Cluster formation was performed using a dependent samples two-sided t-test, with an uncorrected $p$ value threshold of $p < 0.025$ per side, and adjacency was defined as a minimum of two neighboring channels. A permutation distribution was obtained by pooling the averages per participant, irrespective of condition, then randomly assigning them to two

categories using a Monte Carlo simulation. For each of 500 randomizations, t-tests were applied and clusters determined, with the sum of $t$ values of the maximum cluster per randomization as the cluster-based test statistic. The $p$ value was then derived by comparing the uncorrected observed cluster-based test statistic with the permutation distribution and was the proportion of randomizations in which the permuted cluster-based test statistic was larger than the observed cluster-based test statistic. $P$ values smaller than our critical alpha-level (0.025 per side) were deemed significant. Effect size was calculated for the average of the cluster using Cohen's $d = \frac{mean(x_1 - x_2)}{\sqrt{\frac{SD(x_1)^2 + SD(x_2)^2}{2}}}$.

**Regression**. A multiple linear regression was used to test whether clinical factors (right AC–PC coordinates, right total electrical energy delivered, and tremor score) predicted power at the end of training. Power values during DBS-ON within the cluster were averaged over time, frequency, and electrodes.

## Data availability

The numerical source data supporting the findings of this study and underlying the graphs shown are available in the Figshare repository[28]. https://doi.org/10.6084/m9.figshare.25459117.v2.

## Code availability

The code used for these analyses is freely available as a part of the DBSFILT[13] and FieldTrip[14] toolboxes.

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

## Acknowledgements

This work was supported by the Deutsche Forschungsgemeinschaft (DFG) [grant number SW 214/2-1 (CMSR)].

## Author contributions

Study concept and design: C.M.S.R.; Funding acquisition: C.M.S.R.; Study coordination and supervision: C.M.S.R.; Paradigm design: A.V., A.F., C.M.S.R.; Electrode implantation: J.V., L.B.; Clinical management: I.G., J.V., L.B.; Patient recruitment: L.T., N.H., C.M.S.R.; Data collection: A.V., L.T., N.H., C.M.S.R.; Post-operative electrode localization: N.H., M.D., L.B.; Data

analysis: A.V., C.M.S.R.; Interpretation and further analysis: A.V., R.T.K., R.B.I., C.M.S.R.; Further interpretation: A.V., A.F., H.H., S.J.N., A.d.O.A., R.T.K., R.B.I., M.D., L.B., C.M.S.R.; First draft of manuscript: A.V., C.M.S.R.; Extensive further manuscript drafting: A.V., R.T.K., R.B.I., C.M.S.R. Visualization: A.V., L.B.; All authors critically reviewed and approved the final version of the manuscript.

## Funding

## Competing interests
The authors declare no competing interests.
