## [Peer Review File · Communications Biology]

Reviewers' comments:

Reviewer #1 (Remarks to the Author):

Voegtler et al present data showing that VIM stimulation disproportionately improves performance of a learned vs a random sequence, and that there are specific electrophysiological changes that accompany these performance differences. The major behavioral finding is that the DBS ON state improved reaction times for the Learned sequence, but not Random sequences. The major physiological finding is that alpha/beta power is less suppressed in the DBS ON state during performance of learned sequences, but not random sequences. Additional significant findings are that these differences change with practice (Figure 3) and that lead location and energy delivered predicted changes in EEG power.

While the premise is interesting, there are significant methodological issues that need to be addressed. The most important, in my opinion, is that the authors have demonstrated that DBS improves performance of a learned reaction time task, but not necessarily the ability to learn the task. That would require quantifying the improvement with practice in the ON vs OFF states. In addition, the presentation could be significantly improved to make it easier for the reader to follow.

MAJOR ISSUES:

1. If the claim is that VIM DBS improves learning, and not just performance, the authors need to show that task performance improves more quickly in the DBS ON vs OFF state – not just that they're faster in the DBS ON vs OFF state. This would have been facilitated by using different learned sequences in each session (and probably for each patient) but counterbalancing the order should at least partially account for practice effects. In either case, the change in reaction time with practice in the ON vs OFF state needs to be shown, not just the average reaction time across all blocks/sessions. If there is no difference in the rate of decrease in RT as a function of trial number, then in my opinion the authors can claim that there is a performance improvement that is specific to the learned sequence, but not a learning improvement.
2. Even if learning is improved by DBS, are there alternatives to the hypothesis that this learning improvement is caused by the changes in oscillatory power (that is, could these both be true but unrelated)? This possibility should at least be acknowledged. Another possibility is that with tremor reduction, the subjects can concentrate more on the sequence, which may or may not be a reflection of the changes observed in the EEG.
3. The difference between Figure 2 and Figure 3 is unclear. Figure 3 is supposed to be change over time, but the caption says it is DBS-ON minus DBS-OFF for learned vs random trials. Shouldn't the appropriate comparison be DBS ON late minus DBS ON early (and same for DBS OFF)?
4. In panels 2C and 3C, what power is being assessed? Alpha power? Beta power? Alpha and beta power? Broadband power? I think alpha, but I'm not sure.
5. The full regression (scatter plot with line fit) should be shown for the lead location, electrical energy, tremor vs power graphs at least as supplemental figures. It is impossible to assess the validity of a regression without seeing the scatter plots.
6. Were there any patients whose tremor was not well-controlled, or could DBS be tuned to be less effective at tremor suppression but still alter EEG activity? Those could separate effects on tremor control from effects on learning/sequence performance. I don't think the experiments need to be

repeated with "control" stimulation parameters, but it would be nice to know if the effects depend on the level of tremor improvement with DBS.

Minor issues:

1. A panel showing the task design in Figure 1 would be more helpful than the stereotactic X-rays. As it is, the reader has to stop, go to the methods, figure out the task, then come back to the results.
2. "The 144 trials alternated between three cycles in which the stimuli followed a fixed sequence and three cycles in which the stimuli were selected at random, always starting with the sequential cycles." Does this mean that 3 trials were sequential, then 3 trials random, then repeat? In other words, what is a "cycle"?
3. It would make the paper easier to read if n were included in Results text and/or captions so the reader does not have to search for that information in the Methods
4. Panels showing reaction times and power changes as a function of task and DBS state (e.g., 1f, 2c) should show individual data points as a swarm or strip plot so the reader can see if one or a few subjects are driving the results.
5. The specific frequencies the authors consider to be the "alpha" and "beta" bands should be defined.
6. Lines 174-175: "Subcortical, high frequency DBS appears to be able to override disease-related oscillatory patterns, promoting pro-kinetic oscillatory activity." I don't think we can conclude that these oscillatory patterns are "pro-kinetic", especially if tremor is treated.

Reviewer #2 (Remarks to the Author):

This paper reports a finding of improved motor learning of sequences in patients with essential tremor during DBS stimulation of the ventrointermediate thalamic nucleus. Although motor learning improvements in these patients had previously been observed from DBS-VIM stimulation, this study used EEG to identify the neural correlates of these changes in an attempt to elucidate the mechanisms behind this improvement.

The authors observed a changes in alpha/beta oscillations in motor areas in DBS-on and -off conditions, as predicted. They also describe interesting differences in alpha power during early and late learning of motor sequences, as well as some interesting modulations to bilateral motor/parietal networks that may underlie the improvements in motor sequence learning.

In the fourth paragraph of the introduction, the authors describe the changes in alpha/beta oscillations in a confusing way (at least to non-specialists). If they could make clearer what the predictions were in terms of increase/decrease in power or increase/decrease in suppression of alpha and beta, and in what conditions. It might help to outline that first before then reporting the findings.

Reviewer #3 (Remarks to the Author):

Review of “Ventrointermediate thalamic stimulation improves motor learning in humans’ by Voegtle et al., submitted for publication at Communications Biology.

In their study, Voegtle et al., combined VIM-DBS, EEG and a sequence learning task to specify how VIM-DBS is altering cortical oscillations and learning behavior. While the authors made an excellent job in acquiring such difficult data, the manuscript in its current form would benefit from a major revision to improve (1) what was done and (2) why it was done.

In the following, I share with the authors the points that were the most difficult to understand.

1. Intro L66-70 : could you reformulate the precise study predictions ? Because it is currently unclear if you predict a “greater” alpha/beta band suppression or the reverse (a lower beta suppression), and it is also left to the reader to guess whether you had any prediction regarding in which frequency band this cortical power modulation should be observed. It is also weird that the first sentence predict a decrease of alpha/beta power (at rest ?) while the second sentence introduce the reverse prediction (an increase ? “greater cortical modulation” is ambiguous)

2. Methodological introduction (lines 71-79): because the result section have a component during which block 4 and block 1 are contrasted, this means this aspect of the task should be introduced around there (so that the reader would not be lost on lines 111-123). Please also state if this is a novel behavioral task novel or a well-validated task ?

3. Behavior part (line 80-85):

- Line 83: there is a problem of units: I guess % should be replaced by ms. (general advice: please edit and check all the manuscript since in many places the ms. could be improved and clarified).
- The RT result paragraph is hard to understand whereas the figure suggest this is pretty simple. I suggest to report in two separate sentence the results ON vs. OFF DBS to explain where the interaction comes from because current formulation is confusing and ambiguous. Fig 1f suggest that random sequence induced longer RT than learned sequence only ON DBS but not OFF DBS but this intuition is not supported by any statistical data (missing post-hoc testing to find out where the interaction came from). Please rewrite the results to clarify this + add post-hoc statistics instead of t-tests because this inflates type 1 error probability.
- I failed to understand the meaning of the sentence ““At the end of training when the stimuli followed a fixed sequence...”; there is not enough information prior to this sentence to understand its meaning and why it is important to introduce the fact that there was a training part in this experiment (?) and why they begin by reporting results for a “fixed sequence” (does this correspond to the random or to the learned sequence ?); terminology should be homogeneous throughout the manuscript.

4. EEG part (line 86-123):

a) Line 86: ON vs. OFF alpha/beta band suppression: please specify the frequency range of the significant cluster and where is it located (which channels ?) and which trials were used for his first result (which frequency, trials and channels were tested to report a single statistics ???)

b) The next sentence (Line 88) begin by “at the end of training” but we do not know why now stimuli followed a “fixed sequence” : please add the necessary info (we are within block 1 ? before ? during which experimental condition: random or learning condition ?). From fig 2 it appears that we are now focusing on the last block of trials (block 4) but there is no justification provided

a) Line 92: the reported cluster ($t=5660$) is big but we do not know from which sensors it was extracted (average of all electrodes ?) such that the rest of the results are hard to interpret (e.g., same criticism apply for fig 2c: where was this analysis performed, over which electrodes ?); similarly, the same result (“during DBS-ON, alpha/beta power was less suppressed than during DBS-OFF in a cluster spanning a time window of ~320 ms, starting ~580 ms after stimulus onset (cluster- $t = 5660$, $p_{pos} = 0.008$, $SD = 0.004$, Cohen’s $d = 1.779$ ” : could you please explain during which condition (random vs. learned and at which electrodes was this effect was tested?). Without these information, results are hard to evaluate. (if this related for fig 2a, I guess this refers to learning trials but I am not sure from the text)

c) Line 96: “post-hoc evaluation confirmed an interaction (F...): this is confusing: the ANOVA tells you that there is an interaction (F)... and then to decipher this interaction we usually use post-hoc (and specify which one in the text).8; alpha/beta suppression in M1 ipsilateral to movement (is this a new result in the discussion ?)

d) Line 110: Was the lateral and anterior effect of VIM DBS expected (from the known anatomy and physiology ?): this was not introduced before leaving again the reader with no input to judge the relevance of this result (and I think it is not even discussed after)

e) Late vs. early blocks (block 4 vs. 1): please justify why these blocks were compared (I suspect this is to isolate learning effects but this would require the behavior of block 4 and block 1 to be discussed and illustrated; please correct.

Figures: Legend figure 2 c and 3c: for which sensor are the average data from ? grand average EEG ? what would that mean ?

5. Discussion

Line 127-128: “we observed an increase of alpha power”: is this an absolute increase or a lower decrease of alpha ? please find a specific terminology and stick to it throughout the manuscript

Line 165-68: This truism could be removed (the effect of stimulation depends on what we do when stimulated)

In conclusion, I congratulate the authors for their efforts to acquire and analyze a complex and rare

data-set. I would suggest a major revision to clarify their aims and how they were addressed (the result section has to be rewritten)

Reviewer #1 (Remarks to the Author):

Voegtle et al present data showing that VIM stimulation disproportionately improves performance of a learned vs a random sequence, and that there are specific electrophysiological changes that accompany these performance differences. The major behavioral finding is that the DBS ON state improved reaction times for the Learned sequence, but not Random sequences. The major physiological finding is that alpha/beta power is less suppressed in the DBS ON state during performance of learned sequences, but not random sequences. Additional significant findings are that these differences change with practice (Figure 3) and that lead location and energy delivered predicted changes in EEG power.

While the premise is interesting, there are significant methodological issues that need to be addressed. The most important, in my opinion, is that the authors have demonstrated that DBS improves performance of a learned reaction time task, but not necessarily the ability to learn the task. That would require quantifying the improvement with practice in the ON vs OFF states.

In addition, the presentation could be significantly improved to make it easier for the reader to follow.

We thank the Reviewer for their interest in our work and for the helpful suggestions. We have added an assessment of changes in performance over time to address the learning issue and provide a new panel to display these behavioural results. We have also added a panel to depict significant differences in spectral power in the ON vs. OFF states, as well as more detailed figure legends to improve the presentation. We have substantially revised the whole manuscript, adding additional explanation of our approach and findings. We address each of the specific points raised by the Reviewer below.

MAJOR ISSUES:

1. If the claim is that VIM DBS improves learning, and not just performance, the authors need to show that task performance improves more quickly in the DBS ON vs OFF state – not just that they're faster in the DBS ON vs OFF state. This would have been facilitated by using different learned sequences in each session (and probably for each patient) but counterbalancing the order should at least partially account for practice effects. In either case, the change in reaction time with practice in the ON vs OFF state needs to be shown, not just the average reaction time across all blocks/sessions. If there is no difference in the rate of decrease in RT as a function of trial number, then in my opinion the authors can claim that there is a performance improvement that is specific to the learned sequence, but not a learning improvement.

We thank the Reviewer for this important point. We operationalized learning by using a normalized learning score for each block, one that corrects for global changes in performance unrelated to learning (e.g., general improvement or fatigue). We used a repeated measures ANOVA with these data with the factors *Time* (the four blocks), *Stimulation Mode* (ON vs OFF), and *Stimulation Order* (DBS-ON first or second). The Reviewer also suggests that tremor reduction could account for the learning improvement (see below). To address this, we included the change in tremor between DBS-ON and DBS-OFF as well as the total energy

delivered during DBS as covariates in the analysis. The analysis has been added to the Introduction at line 80:

“As our measure of learning, we derived a normalized learning score for each block, comparing reaction times (RTs) for repeated and random sequences within the block. We then examined oscillatory activity recorded during the repeated and random sequences, to investigate whether VIM-DBS has an impact on established oscillatory correlates of motor sequence learning.”

And to the Methods section at line 342:

“To quantify learning, we derived a normalized learning score, defined as the ratio of the difference in mean RT for repeated and random sequences, divided by the mean RT for random sequences. This measure was calculated for each of the four blocks within a session (DBS-ON or DBS-OFF). This provides a way of looking at learning across the four blocks that is independent of general changes in RT which might occur due to changes in the strength of the stimulus–response mapping or fatigue. The normalized learning scores were analyzed with a repeated measures ANOVA with the within-subject factors *Stimulation Mode* (DBS-ON, DBS-OFF) and *Time* (Blocks 1 to 4), and *Stimulation Order* (ON First, OFF First) as a between-subject factor. We included *Tremor Score Difference* (DBS-OFF - DBS-ON) and the *Total Electrical Energy Delivered* as covariates. Post hoc tests are reported following Bonferroni correction for multiple comparisons. 95% confidence intervals (95% CI) are provided. To evaluate whether simple RTs are affected by VIM-DBS, irrespective of learning, we applied a paired T-test to the baseline RTs to the random sequence in Block 1 during DBS-ON compared with DBS-OFF, when no disruption by learning would be expected.”

To the Results section at line 101:

“A three-way interaction was observed between *Stimulation Mode* (DBS-ON, DBS-OFF), *Time* (Block 1 to Block 4), and *Tremor Score Difference* ($F(3,24) = 5.84$, $p = 0.004$; $\eta^2 = 0.42$), and correcting for *Tremor Score Difference*, the two-way interaction between *Stimulation Mode* and *Time* remained significant ($F(3,24) = 3.70$, $p = 0.025$, $\eta^2 = 0.32$; Fig. 1e). The normalized learning score was greater for DBS-ON compared to DBS-OFF in all four blocks. Pairwise post hoc testing showed that the only significant difference between DBS-ON and DBS-OFF was a greater learning score when DBS was on in Block 4 (DBS-ON: $M = 9.59$, 95% CI [3.89 15.3]; DBS-OFF: $M = 4.61$, 95% CI [1.33 7.89]; $p = 0.035$). A two-way interaction was also found between *Tremor Score Difference* and *Time*, with patients with the greatest improvement in tremor score showing the greatest improvement in learning score over time ($F(3,24) = 3.38$, $p = 0.035$, $\eta^2 = 0.30$) but not between *Tremor Score Difference* and *Stimulation Mode*. No significant three-way interactions were found between *Stimulation Mode*, *Time*, and *Stimulation Order*, or between *Stimulation Mode*, *Time*, and *Total Electrical Energy Delivered*. Baseline RTs to the random sequence in Block 1 did not differ according to whether DBS was on ($M = 633.5$, $SD 111.3$) or off ($M = 650.3$, $SD 111.6$; $T(11) = -0.89$, $p = 0.40$).”

Results are presented in Figure 1:

Fig. 1 Illustration of stereotactic targeting of ventrointermediate nucleus of the thalamus (VIM), experimental design, and impact of deep brain stimulation of the VIM on motor sequence learning. **a**. Stereotactic x-ray depicting intraoperative electrode location. Red crosshair: contact in the VIM (Vlpv-Morel²⁵). **b**. Postoperative CT co-registered to preoperative magnetic resonance imaging to establish electrode placement. White circle: CT-artifact of electrode contact in the VIM. Purple line: trajectory through the intracommisural line (IL). Red crosshair: electrode position in co-registered intraoperative x-ray. **c**. Stereotactic atlas of the thalamus²⁶ at IL-level. Red crosshair: represents electrode position in relation to the IL. **d**. Experiment: serial reaction time task, with alternating runs of repeated sequences (S) and random (R) trials, in 4 blocks. **e**. Interaction for the normalized learning score between *Stimulation Mode* (DBS-ON, DBS-OFF) and *Time* (Block 1 to Block 4), showing estimated marginal means and standard errors over individual scores (n = 12).

And to the Discussion at line 169:

“Motor sequence learning was greater when VIM-DBS was on compared to off, with the learning boost during stimulation becoming significant in the final block. RTs in response to the random stimuli did not differ according to whether DBS was on or off, suggesting that DBS-ON enhanced motor sequence learning rather than resulted in a general improvement in motor execution.”

2. Even if learning is improved by DBS, are there alternatives to the hypothesis that this learning improvement is caused by the changes in oscillatory power (that is, could these both be true but unrelated)? This possibility should at least be acknowledged. Another possibility is that with tremor reduction, the subjects can concentrate more on the sequence, which may or may not be a reflection of the changes observed in the EEG.

We thank the Reviewer for these suggestions. The consideration of tremor reduction is addressed in our response to point 1. We also consider a possible effect of tremor reduction on attention. These points are now included in the Discussion at line 234:

“We have suggested that VIM-DBS produces a change in oscillatory power and that these changes contribute to the observed improvement in learning. We recognize there are other, less direct ways in which these changes might come about from VIM-DBS. For example, with the tremor reduction observed during VIM-DBS, the patients might have more attentional processes to devote to extracting the stimulus pattern during the sequence blocks. On the other hand, although the regression indicated some relationship between oscillatory power and tremor score, the latter did not fully account for the difference in mean learning score with DBS-ON compared with DBS-OFF, suggesting that neither general motor improvement or changes in attentional state fully explain the improved motor sequence learning. We note that while attention improvement could theoretically accompany tremor amelioration, we did not directly test attention here.”

3. The difference between Figure 2 and Figure 3 is unclear. Figure 3 is supposed to be change over time, but the caption says it is DBS-ON minus DBS-OFF for learned vs random trials. Shouldn't the appropriate comparison be DBS ON late minus DBS ON early (and same for DBS OFF)?

Thank you for pointing out the confusion in our presentation. The Reviewer is correct that late and early measures should be contrasted. We did this and revised our description of what is now Figure 4 and intext to make our approach clearer. To examine the effect time, we created the contrast between Block 4 and Block 1 during the repeated sequence and compared the DBS-ON against DBS-OFF conditions. Similarly, we created a contrast between Block 4 and Block 1 during the random sequence and compared DBS-ON against DBS-OFF. We predicted changes in the contrast for the repeated sequences, as performance improved, while not necessarily during random trials. Again, our primary interest was in how those changes are affected by DBS, hence we compared both stimulation conditions (DBS-ON, DBS-OFF) against each other.

We have added the following at line 92 of the Introduction:

“To account for the effects of time, we contrasted trials from Block 4 to Block 1 separately for the repeated and the random sequences, and compared the DBS-ON against DBS-OFF conditions.”

We have changed the following at line 365 of the Methods:

“To account for the ~~time evolution of effects of learning over~~ time on oscillatory power, we performed a third analysis, ~~testing comparing the contrasted trials between~~ of Block 4 to Block 1 ~~during the learned sequence and the random trials, and compared the between~~ DBS-ON ~~against~~ and DBS-OFF conditions, for repeated ~~and random~~ trials. We performed an analogous analysis for random trials.”

The caption for Fig. 4 now specifies:

Fig. 4 Changes in oscillatory power over the course of **time responding to a sequence** (difference between Block 4 and Block 1; $n = 12$). **a** Grand-average difference between DBS-ON and DBS-OFF for **the contrast between Block 4 and Block 1 during repeated** sequences. **b** Grand-average difference between DBS-ON and DBS-OFF for **the contrast between Block 4 and Block 1 during random** trials. **c** Power values within the determined cluster were averaged over time, frequency, and electrodes for both stimulation modes and sequence types. Interaction for power values between *Stimulation Mode* and *Sequence Type*, showing means and standard errors. **d** Location of the observed cluster, representing significant difference between DBS-ON and DBS-OFF **for the contrast over time** during **repeated** sequences ($\text{cluster-}t = 7237$, $p_{\text{pos}} = 0.012$, $SD = 0.005$, $\text{Cohen's } d = 2.485$). The cluster included electrodes spanning the contralateral cortex over a central-parietal, parietal-occipital region. The channel contributing most to the cluster was CP5. **e** Left panel: grand-average of channel CP5 during DBS-ON; right panel: grand-average of channel CP5 during DBS-OFF. **a, b, d**: x-axis = Time: -200 to 1200 ms; y-axis = Frequency: 2-30 Hz.

4. In panels 2C and 3C, what power is being assessed? Alpha power? Beta power? Alpha and beta power? Broadband power? I think alpha, but I'm not sure.

In both panels, we assessed the power within the cluster determined by the cluster-based permutation test. We hypothesized that power would differ in the alpha and/or beta frequency ranges. To avoid pre-defining frequency band cut-offs, we examined the frequency range from 2-30 Hz. Cluster-based permutation tests addressed the comparisons made in multiple frequencies, enabling identification of frequencies at which power differed without making a priori assumptions about cut-off frequencies. We have added the following information to both legends of Figure 3 and 4 (lines 414 and 425):

“Power values within the determined cluster were averaged over time, frequency, and electrodes for both Stimulation Modes and Sequence Types.”

5. The full regression (scatter plot with line fit) should be shown for the lead location, electrical energy, tremor vs power graphs at least as supplemental figures. It is impossible to assess the validity of a regression without seeing the scatter plots.

We have added Supplementary Figure 1 to visualize partial regression plots, providing graphical information about the contribution of each predictor variable to the model.

6. Were there any patients whose tremor was not well-controlled, or could DBS be tuned to be less effective at tremor suppression but still alter EEG activity? Those could separate effects on tremor control from effects on learning/sequence performance. I don't think the experiments need to be repeated with "control" stimulation parameters, but it would be nice to know if the effects depend on the level of tremor improvement with DBS.

All patients experienced improvement in tremor score with DBS-ON compared with DBS-OFF. We have added the following at line 116 in the first paragraph of the Results:

“All patients showed a reduction in tremor when DBS stimulation was on compared to when it was off.”

The interaction between *Stimulation Mode* (DBS-ON vs. DBS-OFF) and *Time* now includes the covariate *Tremor Score Difference* (see responses to points 1 and 2 above), consistent with the proposal that motor sequence learning, analogous to tremor suppression, is achieved through disruption of pathological activity in the tremor network by VIM-DBS. However, no significant interaction was observed between *Stimulation Mode* (DBS-ON vs. DBS-OFF) and *Tremor Score Difference*, suggesting that improvement in tremor alone cannot completely explain the improvement in motor sequence learning with DBS-ON. We have also added the following at line 173 to the first paragraph of the Discussion:

“The absence of an interaction between *Tremor Score Difference* and *Stimulation Mode* suggests that change in tremor score does not fully account for the motor learning score improvement.”

However, the regression analysis showed that the mean power with DBS-ON was correlated with the tremor rating score during stimulation, showing that these measures are not entirely independent. We have also added the following to the Discussion at line 239:

“On the other hand, although the regression indicated some relationship between oscillatory power and tremor score, the latter did not fully account for the difference in mean learning score with DBS-ON compared with DBS-OFF, suggesting that neither general motor improvement or changes in attentional state fully explain the improved motor sequence learning. We note that while attention improvement could theoretically accompany tremor amelioration, we did not directly test attention here.”

Minor issues:

1. A panel showing the task design in Figure 1 would be more helpful than the stereotactic X-rays. As it is, the reader has to stop, go to the methods, figure out the task, then come back to the results.

We agree and have added the task design to Figure 1.

Fig. 1 Illustration of stereotactic targeting of ventrointermediate nucleus of the thalamus (VIM), experimental design, and impact of deep brain stimulation of the VIM on motor sequence learning. **a.** Stereotactic x-ray depicting intraoperative electrode location. Red crosshair: contact in the VIM (Vlpv-Morel²⁵). **b.** Postoperative CT co-registered to preoperative magnetic resonance imaging to establish electrode placement. White circle: CT-artifact of electrode contact in the VIM. Purple line: trajectory through the intracommissural line (IL). Red crosshair: electrode position in co-registered intraoperative x-ray. **c.** Stereotactic atlas of the thalamus²⁶ at IL-level. Red crosshair: represents electrode position in relation to the IL. **d.** Experiment: serial reaction time task, with alternating runs of repeated sequences(S) and random (R) trials, in 4 blocks. **e.** Interaction for the normalized learning score between *Stimulation Mode* (DBS-ON, DBS-OFF) and *Time* (Block 1 to Block 4), showing estimated marginal means and standard errors over individual scores (n = 12).

2. “The 144 trials alternated between three cycles in which the stimuli followed a fixed sequence and three cycles in which the stimuli were selected at random, always starting with the sequential cycles.” Does this mean that 3 trials were sequential, then 3 trials random, then repeat? In other words, what is a “cycle”?

In addition to the task design added to Figure 1, we adjusted the text in the Methods at line 285 to the following, to make our intentions clearer:

“The 144 trials alternated **twice** between three repetitions of the 12-item sequence (36 trials) **eyes in which the stimuli followed a fixed sequence** and 36 random trials **three eyes in which the stimuli were selected at random**, always starting with the sequential trials **eyes** (Fig 1d).”

3. It would make the paper easier to read if n were included in Results text and/or captions so the reader does not have to search for that information in the Methods

We have added n to Figures 1, 2, 3, and 4 and included it at the beginning of the Results at line 101:

“The data from 12 participants were included in all analyses.”

4. Panels showing reaction times and power changes as a function of task and DBS state (e.g., 1f, 2c) should show individual data points as a swarm or strip plot so the reader can see if one or a few subjects are driving the results.

We included individual data points to all relevant subpanels in Figures 1, 3, and 4.

5. The specific frequencies the authors consider to be the “alpha” and “beta” bands should be defined.

We included the following in the Introduction at line 63:

“Modulation of oscillatory brain activity has been observed in the alpha (8-12 Hz)^{5,10,11} and beta bands (13-30 Hz)¹² during motor sequence learning.”

6. Lines 174-175: “Subcortical, high frequency DBS appears to be able to override disease-related oscillatory patterns, promoting pro-kinetic oscillatory activity.” I don’t think we can conclude that these oscillatory patterns are “pro-kinetic”, especially if tremor is treated.

We agree and have removed this from the Discussion at line 250:

“Subcortical, high frequency DBS appears to override disease-related oscillatory patterns; ~~promoting pro-kinetic oscillatory activity.~~”

Reviewer #2 (Remarks to the Author):

This paper reports a finding of improved motor learning of sequences in patients with essential tremor during DBS stimulation of the ventrointermediate thalamic nucleus. Although motor learning improvements in these patients had previously been observed from DBS-VIM stimulation, this study used EEG to identify the neural correlates of these changes in an attempt to elucidate the mechanisms behind this improvement.

The authors observed a changes in alpha/beta oscillations in motor areas in DBS-on and -off conditions, as predicted. They also describe interesting differences in alpha power during early and late learning of motor sequences, as well as some interesting modulations to bilateral motor/parietal networks that may underlie the improvements in motor sequence learning.

We thank the Reviewer for their interest in our work.

In the fourth paragraph of the introduction, the authors describe the changes in alpha/beta

oscillations in a confusing way (at least to non-specialists). If they could make clearer what the predictions were in terms of increase/decrease in power or increase/decrease in suppression of alpha and beta, and in what conditions. It might help to outline that first before then reporting the findings.

Thank you for this useful suggestion. We agree that more details are needed to make the direction of change clearer. At the end of the first paragraph of the Introduction at line 64, we now explain the distinction between a reduction in suppression and an increase in power as follows:

“For example, patients with Parkinson’s disease ~~have show~~ enhanced and prolonged beta band ~~power suppression decrease compared to controls~~. This enhanced suppression ~~that~~ correlated with impairments in motor sequence learning¹². ET patients also ~~exhibit increased enhanced~~ suppression in the alpha/beta band during movement, ~~similar comparable~~ to Parkinson’s disease patients⁶. We hypothesized that oscillatory power is modulated by VIM-DBS in patients with ET, ~~with-including a reduction of pathological electrophysiological brain activity in the alpha/beta frequency-band power suppression~~. Note that a reduced suppression of alpha/beta power is distinct from an increase in power. Given that sequence learning is modulated by VIM-DBS⁴, we predicted ~~that a reduction in alpha/beta band power suppression cortical power modulation would accompany improved performance of a repeated sequence of finger movements compared to when they produced a series of random finger movements.~~”

And the following at line 178 of the Discussion:

“Note that, in contrast to motor cortical regions, movement-related suppression of an ongoing alpha/beta rhythm has not been reported in the parietal cortex, and we therefore refer to a power increase.”

Reviewer #3 (Remarks to the Author):

Review of “Ventrointermediate thalamic stimulation improves motor learning in humans” by Voegtle et al., submitted for publication at Communications Biology.

In their study, Voegtle et al., combined VIM-DBS, EEG and a sequence learning task to specify how VIM-DBS is altering cortical oscillations and learning behavior. While the authors made an excellent job in acquiring such difficult data, the manuscript in its current form would benefit from a major revision to improve (1) what was done and (2) why it was done.

We thank the Reviewer for the positive evaluation of our study and for the detailed recommendations, to which we have responded point by point below, substantially revising the manuscript.

In the following, I share with the authors the points that were the most difficult to understand.

1. Intro L66-70 : could you reformulate the precise study predictions ? Because it is currently unclear if you predict a “greater” alpha/beta band suppression or the reverse (a lower beta suppression), and it is also left to the reader to guess whether you had any prediction regarding in which frequency band this cortical power modulation should be observed. It is

also weird that the first sentence predict a decrease of alpha/beta power (at rest ?) while the second sentence introduce the reverse prediction (an increase ? “greater cortical modulation” is ambiguous).

Thank you for pointing out the lack of clarity regarding direction of change, a point that was also made by Reviewer 2. We have added further details regarding reduced suppression and increased power and made our predictions clearer at the end of the first paragraph of the Introduction at line 64 as follows:

“For example, patients with Parkinson’s disease ~~have show~~ enhanced and prolonged beta band ~~power suppression decrease compared to controls. This enhanced suppression that~~ correlated with impairments in motor sequence learning¹². ET patients also ~~exhibit increased enhanced~~ suppression in the alpha/beta band during movement, ~~similar comparable~~ to Parkinson’s disease patients⁶. We hypothesized that oscillatory power is modulated by VIM-DBS in patients with ET, ~~with including a reduction of pathological electrophysiological brain activity in the alpha/beta frequency-band power suppression. Note that a reduced suppression of alpha/beta power is distinct from an increase in power.~~ Given that sequence learning is modulated by VIM-DBS⁴, we predicted that there would be a ~~reduction in alpha/beta band power suppression cortical power modulation~~ while patients performed a repeated sequence of finger movements ~~compared to when they produced a series of random finger movements.”~~

And again, in the discussion at line 177:

“...we observed an increase in alpha power in a cluster centered over left parietal cortex, contralateral to the moving hand. ~~Note that, in contrast to motor cortical regions, movement-related suppression of an ongoing alpha/beta rhythm is not established for the parietal cortex.”~~

2. Methodological introduction (lines 71-79): because the result section have a component during which block 4 and block 1 are contrasted, this means this aspect of the task should be introduced around there (so that the reader would not be lost on lines 111-123). Please also state if this is a novel behavioral task novel or a well-validated task?

We added the following information about blocks regarding the behavioral analysis in the Introduction starting at line 80:

“As our measure of learning, we derived a normalized learning score for each block, comparing reaction times (RTs) for repeated and random sequences within the block.”

And for the electrophysiological analysis (line 92):

“To account for the effects of time, we contrasted trials from Block 4 to Block 1 separately for the repeated and the random sequences, and compared the DBS-ON against DBS-OFF conditions.”

Additionally, we have included the task design in Figure 1.

Fig. 1 Illustration of stereotactic targeting of ventrointermediate nucleus of the thalamus (VIM), experimental design, and impact of deep brain stimulation of the VIM on motor sequence learning. **a.** Stereotactic x-ray depicting intraoperative electrode location. Red crosshair: contact in the VIM (Vlpv-Morel²⁵). **b.** Postoperative CT co-registered to preoperative magnetic resonance imaging to establish electrode placement. White circle: CT-artifact of electrode contact in the VIM. Purple line: trajectory through the intracommissural line (IL). Red crosshair: electrode position in co-registered intraoperative x-ray. **c.** Stereotactic atlas of the thalamus²⁶ at IL-level. Red crosshair: represents electrode position in relation to the IL. **d.** Experiment: serial reaction time task, with alternating runs of repeated sequences (S) and random (R) trials, in 4 blocks. **e.** Interaction for the normalized learning score between *Stimulation Mode* (DBS-ON, DBS-OFF) and *Time* (Block 1 to Block 4), showing estimated marginal means and standard errors over individual scores (n = 12).

We have also included that the task is well-established, citing studies using the task in line 60 of the Introduction:

“Here we investigate the impact of VIM-DBS on oscillatory activity in cortical nodes of the motor learning network using the **well-established** serial reaction time task (SRTT)^{5,9-12}.”

3. Behavior part (line 80-85):

a) Line 83: there is a problem of units: I guess % should be replaced by ms. (general advice: please edit and check all the manuscript since in many places the ms. could be improved and clarified).

Thank you for pointing out this issue. The % actually refers to the 95% confidence interval. To make this clearer, we have added the definition of CI to line 352 as follows:

“95% confidence intervals (95% CI) are provided.”

b) The RT result paragraph is hard to understand whereas the figure suggest this is pretty simple. I suggest to report in two separate sentence the results ON vs. OFF DBS to explain where the interaction comes from because current formulation is confusing and ambiguous. Fig 1f suggest that random sequence induced longer RT than learned sequence only ON DBS but not OFF DBS but this intuition is not supported by any statistical data (missing post-hoc testing to find out where the interaction came from). Please rewrite the results to clarify this + add post-hoc statistics instead of t-tests because this inflates type 1 error probability.

Reviewer 1 had similar concerns, and we have now completely revised this section. We now calculate a mean normalized learning score, which quantifies the difference between responding to repeated and random sequences, for each Block. We also now report the post hoc statistics for the interaction instead of T-tests. The revised section is in the Results section at line 101:

“A three-way interaction was observed between *Stimulation Mode* (DBS-ON, DBS-OFF), *Time* (Block 1 to Block 4), and *Tremor Score Difference* ($F(3,24) = 5.84, p = 0.004; \eta^2 = 0.42$), and correcting for *Tremor Score Difference*, the two-way interaction between *Stimulation Mode* and *Time* remained significant ($F(3,24) = 3.70, p = 0.025, \eta^2 = 0.32$; Fig. 1e). The normalized learning score was greater for DBS-ON compared to DBS-OFF in all four blocks. Pairwise post hoc testing showed that the only significant difference between DBS-ON and DBS-OFF was a greater learning score when DBS was on in Block 4 (DBS-ON: $M = 9.59, 95\% \text{ CI } [3.89 \text{ } 15.3]$; DBS-OFF: $M = 4.61, 95\% \text{ CI } [1.33 \text{ } 7.89]$; $p = 0.035$). A two-way interaction was also found between *Tremor Score Difference* and *Time*, with patients with the greatest improvement in tremor score showing the greatest improvement in learning score over time ($F(3,24) = 3.38, p = 0.035, \eta^2 = 0.30$) but not between *Tremor Score Difference* and *Stimulation Mode*. No significant three-way interactions were found between *Stimulation Mode*, *Time*, and *Stimulation Order*, or between *Stimulation Mode*, *Time*, and *Total Electrical Energy Delivered*. Baseline RTs to the random sequence in Block 1 did not differ according to whether DBS was on ($M = 633.5, \text{SD } 111.3$) or off ($M = 650.3, \text{SD } 111.6; T(11) = -0.89, p = 0.40$).”

c) I failed to understand the meaning of the sentence ““At the end of training when the stimuli followed a fixed sequence...””; there is not enough information prior to this sentence to understand its meaning and why it is important to introduce the fact that there was a training part in this experiment (?) and why they begin by reporting results for a “fixed sequence” (does this correspond to the random or to the learned sequence ?); terminology should be homogeneous throughout the manuscript.

We agree that we could do a better job in using consistent terminology. We now use the phrase “repeated sequence” to refer to the condition in which the stimuli follow a fixed sequence, allowing us to refer to the contrast of repeated vs random. In that way we make no assumption that the repeated sequence will be learned/acquired. Having defined the term “training”, we have now replaced “end of acquisition” with “end of training” throughout.

We have added the following clarification at line 288 in the Methods:

“We refer to Block 4 of the experiment as the end of training.”

We have also referred to the term at first mention in the Introduction at line 90:

“at the end of training (Block 4)”

4. EEG part (line 86-123):

a) Line 86: ON vs. OFF alpha/beta band suppression: please specify the frequency range of the significant cluster and where is it located (which channels ?) and which trials were used for his first result (which frequency, trials and channels were tested to report a single statistics ???)

We agree that more information is needed regarding the results of this first test and added the following description to the Results at line 124:

“Turning to the physiological data, when we collapsed across the repeated sequence and random trials at the end of training (Block 4), we observed a significant difference between DBS-ON and DBS-OFF. The cluster was observed over bilateral sensorimotor areas (cluster- $t = 9712$, $p_{\text{pos}} = 0.004$, $SD = 0.003$, Cohen’s $d = 2.156$), spanning a time window of ~500 ms, starting ~500 ms after stimulus onset in the alpha/beta range (~6-16 Hz). The cluster indicated a widespread, bilateral ~~reduction~~ reduced suppression in alpha/beta power ~~suppression~~ when VIM-DBS was on compared with off (Fig. 2).”

We also included a revised Figure (now Figure 2), adding the corresponding t-map to visualize the cluster. We also added the following general information about the data used for the cluster-based permutation tests to the Introduction at line 86:

“We used cluster-based permutation tests¹⁴ to examine oscillatory power from 59 electrodes over an epoch starting 200 ms before to 1200 ms after stimulus onset, with the analysis spanning a frequency range of 2-30 Hz.”

And specified the requested information in the Methods at line 358:

“...we performed non-parametric cluster-based permutation tests¹⁴, including all 59 channels, time points (-200 ms before to 1200 ms after stimulus onset), and frequencies (2-30 Hz). The first analysis was applied to all trials at the end of training-acquisition, including ~~when the sequence was learned and when it was repeated and~~ random trials.”

We added the following at line 190 of the Discussion:

“When examining the data pooled across repeated and random trials, VIM-DBS ~~-ON~~ compared with ~~-OFF~~ was associated with a widespread, bilateral reduced suppression in alpha/beta power, compared to when the stimulation was off. This general effect of VIM-DBS on cortical oscillatory activity appears to have a specific impact on motor sequence learning performance, given the effect on the normalized learning score.”

b) The next sentence (Line 88) begin by “at the end of training” but we do not know why now stimuli followed a “fixed sequence” : please add the necessary info (we are within block 1 ? before ? during which experimental condition: random or learning condition ?). From fig 2 it appears that we are now focusing on the last block of trials (block 4) but there is no justification provided

We now consistently use the term “end of training”, indicating Block 4, as described above.

We have now included a behavioural analysis including the factor *Time* (Block 1 to Block 4). This analysis showed that learning (the RT advantage on sequence trials compared to random trials) was significant at the end of training. This motivated us to concentrate our electrophysiological analysis on the end of training. We adjusted the following paragraph in the Introduction starting at line 88:

“Based on the behavioral results, we examined the general effects of DBS across all trials (repeated and random), ~~comparing DBS-ON with DBS-OFF~~ at the end of training (Block 4). We then performed separate analyses for repeated and random trials, contrasting the ON-OFF conditions at the end of ~~acquisition-training~~.”

c) Line 92: the reported cluster ($t=5660$) is big but we do not know from which sensors it was extracted (average of all electrodes ?) such that the rest of the results are hard to interpret (e.g., same criticism apply for fig 2c: where was this analysis performed, over which electrodes ?); similarly, the same result (“during DBS-ON, alpha/beta power was less suppressed than during DBS-OFF in a cluster spanning a time window of ~320 ms, starting ~580 ms after stimulus onset (cluster- $t = 5660$, $p_{pos} = 0.008$, $SD = 0.004$, Cohen’s $d = 1.779$)” : could you please explain during which condition (random vs. learned and at which electrodes was this effect was tested?). Without these information, results are hard to evaluate. (if this related for fig 2a, I guess this refers to learning trials but I am not sure from the text)

We have now included the following description in the Introduction at line 86:

“We used cluster-based permutation tests¹⁴ to examine oscillatory spectral power from 59 electrodes over an epoch starting 200 ms before to 1200 ms after stimulus onset), with the analysis spanning a frequency range of 2-30 Hz. Based on the behavioral results, we examined the general effects of DBS across all trials (repeated and random), ~~comparing DBS-ON with DBS-OFF~~ at the end of training (Block 4). We then performed separate analyses for repeated and random trials, contrasting the ON-OFF conditions at the end of ~~acquisition training~~.”

And the following in the Results section at line 124:

“Turning to the physiological data, when we collapsed across the repeated sequence and random trials at the end of training (Block 4), we observed a significant difference between DBS-ON and DBS-OFF. The cluster was observed over bilateral sensorimotor areas (cluster- $t = 9712$, $p_{pos} = 0.004$, $SD = 0.003$, Cohen’s $d = 2.156$), spanning a time window of ~500 ms, starting ~500 ms after stimulus onset in the alpha/beta range (~6-16 Hz). The cluster indicated a widespread, bilateral ~~reduction-reduced suppression~~ in alpha/beta power ~~suppression~~ when VIM-DBS was on compared with off (Fig. 2).

At the end of training, ~~when the stimuli followed a fixed repeated sequence, power differences-a difference~~ between DBS-ON and DBS-OFF was observed for the repeated sequence, ~~corresponding with a~~ cluster encompassing the alpha/beta frequency bands. These effects were found over the central and ipsilateral motor area, including M1 and premotor

cortex (PMC; Fig. 3), spanning a time window of ~320 ms, starting ~580 ms after stimulus onset (cluster-t = 5660, $p_{\text{pos}} = 0.008$, SD = 0.004, Cohen's d = 1.779).”

We also included the new Figure 2 to visualize the reported results.

Fig. 2 General impact of stimulation at the end of training (n = 12). **a.** Grand-average during DBS-ON **b.** Grand-average during DBS-OFF **c.** Location of the observed cluster, when testing DBS-ON against DBS-OFF over all trials (cluster-t = 9712, $p_{\text{pos}} = 0.004$, SD = 0.003, Cohen's d = 2.156). The cluster included electrodes spanning the bilateral sensorimotor areas. **a, b, c:** x-axis = Time: -200 to 1200 ms; y-axis = Frequency: 2-30 Hz.

We added the cluster statistics to the corresponding t-map legends in Figures 2, 3, and 4 (lines 410, 418, and 429).

d) Line 96: “post-hoc evaluation confirmed an interaction (F...): this is confusing: the ANOVA tells you that there is an interaction (F...) and then to decipher this interaction we usually use post-hoc (and specify which one in the text).8; alpha/beta suppression in M1 ipsilateral to movement (is this a new result in the discussion ?)

In hindsight, we see that the presentation of these analyses was confusing. We have revised things as follows. We first test our primary hypothesis, which is now stated more clearly at line 72 of the Introduction:

“Given that sequence learning is modulated by VIM-DBS⁴, we predicted **greater reduction in alpha/beta band power suppression cortical power modulation** while patients performed a **repeated** sequence of finger movements **compared to when they produced a series of random finger movements.**”

We then performed a repeated measures ANOVA and observed an interaction. The interaction was assessed with post hoc tests, now described for the end of training analysis in the Results section at line 139:

“~~Post-hoc evaluation confirmed~~ A repeated measures ANOVA showed an interaction between *Stimulation Mode* (DBS-ON, DBS-OFF) and *Sequence Type* (**Repeated**, Random) ($F(1,11) = 9.84$, $p = 0.009$, $\eta^2 = 0.47$). **Post hoc tests showed** that the power difference during **repeated** sequences compared to random sequences was greater with DBS-ON (**Repeated**: $M = 0.22$, 95% CI [-0.18, 0.62]; **Random**: $M = -0.38$, 95% CI [-0.74, -0.02]) compared to DBS-OFF (**Repeated**: $M = -0.95$, 95% CI [-1.39, -0.51]; **Random**: $M = -0.69$, 95% CI [-1.19, -0.20]) ($T(11) = 3.14$, $p = 0.009$, Cohen's d = 0.91) (Fig. 3c).”

e) Line 110: Was the lateral and anterior effect of VIM DBS expected (from the known anatomy and physiology?): this was not introduced before leaving again the reader with no input to judge the relevance of this result (and I think it is not even discussed after)

The evaluation of the electrode location was based on our previous findings (Terzic et al., 2022). We now clarify this in the Introduction at line 95:

“Based on previous findings of a greater impact on motor sequence learning when VIM electrodes were located more laterally⁴, we assessed whether the modulation of oscillatory power by VIM-DBS during sequence learning was dependent on the specific electrode location within the VIM. In addition, we evaluated whether power modulation was related to the degree of tremor amelioration or the total energy delivered by DBS.”

We discussed the findings at line 194 as follows:

“The effect on oscillatory power was greater when the stimulation site was at more lateral aspects of the VIM. ~~regions that are anatomically linked. There are cytoarchitectural and connectivity differences between lateral and medial aspects of VIM, with lateral VIM receiving a large input from cerebellar afferents anatomically linked with cortical and subcortical motor structures^{17,18}. and where~~ Further evidence that lateral VIM is part of a motor network comes from studies ~~single-cell recordings~~ showing tactile and kinesthetic responses in lateral but not the medial VIM during movement^{19,20}. Moreover, we previously observed enhanced motor sequence learning performance with VIM-DBS delivery to more lateral VIM sites compared to more medial sites⁴.”

f) Late vs. early blocks (block 4 vs. 1): please justify why these blocks were compared (I suspect this is to isolate learning effects but this would require the behavior of block 4 and block 1 to be discussed and illustrated; please correct.

The Reviewer is correct that we used a contrast between Block 4 and Block 1, since we expected this contrast would show the largest learning effect. To statistically justify this contrast, we now include *Time*, operationalized as the four training blocks in our behavioral analysis of the mean normalized learning data. While this measure was numerically greater with DBS-ON than DBS-OFF in all four Blocks, the difference only reached significance in Block 4. We then examined how oscillatory power differed in this block between DBS-ON and DBS-OFF, and how the power difference between Blocks 4 and 1 differed between DBS-ON and DBS-OFF.

The behavioral analysis including the factor *Time* has been added to the Methods section at line 342:

“To quantify learning, we derived a normalized learning score, defined as the ratio of the difference in mean RT for repeated and random sequences, divided by the mean RT for random sequences. This measure was calculated for each of the four blocks within a session (DBS-ON or DBS-OFF). It provides a way of looking at learning across the four blocks independent of general changes in RT that might occur due to changes in the strength of the stimulus-response mapping or fatigue. The normalized learning scores were analyzed with a repeated measures ANOVA with the within-subject factors *Stimulation Mode* (DBS-ON, DBS-OFF) and *Time*

(Blocks 1 to 4), and *Stimulation Order* (ON First, OFF First) as a between-subject factor. We included *Tremor Score Difference* (DBS-OFF - DBS-ON) and the *Total Electrical Energy Delivered* as covariates. Post hoc tests are reported following Bonferroni correction for multiple comparisons.”

The findings are in the Results section at line 101:

“A three-way interaction was observed between *Stimulation Mode* (DBS-ON, DBS-OFF), *Time* (Block 1 to Block 4), and *Tremor Score Difference* ($F(3,24) = 5.84, p = 0.004; \eta^2 = 0.42$), and correcting for *Tremor Score Difference*, the two-way interaction between *Stimulation Mode* and *Time* remained significant ($F(3,24) = 3.70, p = 0.025, \eta^2 = 0.32$; Fig. 1e). The normalized learning score was greater for DBS-ON compared to DBS-OFF in all four blocks. Pairwise post hoc testing showed that the only significant difference between DBS-ON and DBS-OFF was a greater learning score when DBS was on in Block 4 (DBS-ON: $M = 9.59, 95\% \text{ CI } [3.89 \text{ } 15.3]$; DBS-OFF: $M = 4.61, 95\% \text{ CI } [1.33 \text{ } 7.89]$; $p = 0.035$). A two-way interaction was also found between *Tremor Score Difference* and *Time*, with patients with the greatest improvement in tremor score showing the greatest improvement in learning score over time ($F(3,24) = 3.38, p = 0.035, \eta^2 = 0.30$) but not between *Tremor Score Difference* and *Stimulation Mode*. No significant three-way interactions were found between *Stimulation Mode*, *Time*, and *Stimulation Order*, or between *Stimulation Mode*, *Time*, and *Total Electrical Energy Delivered*.”

Figures: Legend figure 2 c and 3c: for which sensor are the average data from ? grand average EEG ? what would that mean ?

In both panels, we assessed the power within the cluster determined by the cluster-based permutation test. We hypothesized that power would differ in the alpha and/or beta frequency ranges. To avoid pre-defining frequency bands, however, we examined the frequency range from 2-30 Hz. Using cluster-based permutation tests addressed the comparisons in multiple frequencies. We have added the following information to both legends of Figure 3 and 4 (lines 414 and 425):

“Power values within the determined cluster were averaged over time, frequency, and electrodes for both stimulation modes and stimulus conditions.”

5. Discussion

Line 127-128: “we observed an increase of alpha power”: is this an absolute increase or a lower decrease of alpha ? please find a specific terminology and stick to it throughout the manuscript

To clarify the distinction between suppression of power and the absolute increase in alpha, we added the following to the Introduction at line 71:

“Note that a reduction in alpha/beta suppression is distinct from an increase in power.”

And the Discussion at line 177:

“...we observed an increase in alpha power in a cluster centered over left parietal cortex, contralateral to the moving hand. Note that, in contrast to motor cortical regions, movement-

related suppression of an ongoing alpha/beta rhythm has not been reported in the parietal cortex.”

Line 165-68: This truism could be removed (the effect of stimulation depends on what we do when stimulated)

We agree and have removed this from the discussion.

In conclusion, I congratulate the authors for their efforts to acquire and analyze a complex and rare data-set. I would suggest a major revision to clarify their aims and how they were addressed (the result section has to be rewritten)

We thank the Reviewer for the encouraging words and for the many helpful suggestions.

Reviewers' comments:

Reviewer #1 (Remarks to the Author):

The authors made several revisions that addressed some of my previous questions, but also raise new ones. Overall, there does seem to be a small but significant effect of VIM DBS on sequence learning beyond a general improvement in movement, accompanied by differences in brain oscillations between DBS on/off and repeated vs random sequences. My remaining comments mostly center around presenting the "learning score" more clearly, and the discussion surrounding VIM anatomy.

Major points:

1. The normalized learning score should be defined at least briefly in the Results before describing outcomes.
2. Please clarify the definition of the learning score, preferably with an equation. It is defined "...as the ratio of the difference in mean RT for repeated and random sequences, divided by the mean RT for random sequences". This is confusing. I think this means $(RT_{repeated} - RT_{random}) / RT_{random}$. However, "the ratio of the difference divided by..." sounds like there are 2 quotients, so I'm not sure what was calculated. If my definition is correct, doesn't that mean that the DBS on condition was associated with longer RT relative to the random condition across blocks (Fig. 1E)? This needs to be clearly defined since the order in which the difference is taken gives opposite interpretations.
3. Line 198-200. "Further evidence that lateral VIM is part of a motor network comes from studies showing tactile and kinesthetic responses in lateral VIM but not the medial VIM during movement". Here, the authors cite a paper describing the lack of kinesthetic/tactile responses in medial VIM, arguing that lateral VIM is part of a motor network while medial VIM is not. However, at least in citation 20, tactile/kinesthetic responses were only tested in the contralateral upper limb. Since there is a face \rightarrow arm \rightarrow leg somatotopy to VIM moving medial to lateral, I think it is incorrect to draw the conclusion that lateral VIM is part of a motor network but medial VIM is not. In fact, it seems the strongest evidence that VIM is part of a motor network is the fact that VIM DBS improves tremor.
4. Lines 195-196. Similarly, the authors argue that lateral VIM receives more cerebellar afferents than medial VIM. "There are cytoarchitectural and connectivity differences between lateral and medial aspects of VIM, with lateral VIM receiving a large input from cerebellar afferents." I don't think this is true. The authors cite two old nonhuman primate papers, but there are many more recent papers in humans using tractography that would be more directly relevant. Generally, VIM thalamus is considered to receive cerebellar afferents, almost by definition. Perhaps part of the problem is that the nomenclature and definitions for thalamic subregions are inconsistent (sometimes based on cytoarchitecture, sometimes based on afferents). For the current paper, the relevant question is whether the medial vs lateral lead locations have different afferents. That seems unlikely since DBS effectively suppressed tremor in all the patients, suggesting that the leads were all in locations with dense cerebellar afferents.

Minor points:

1. Would rephrase "mean normalized learning score" on line 76, or describe it as a measure of learning since at this point in the manuscript the reader won't know what this means.
2. I think lines 76-94 are too much detail for the introduction, and moving them later would improve readability. However, I do not feel strongly about this.
3. Line 169: It's not clear what "Motor sequence learning was greater..." means. Maybe "faster" or "more efficient"?

Reviewer #3 (Remarks to the Author):

The authors have addressed all my concerns. Congratulations for achieving such a difficult study in

relatively rare patients.

Reviewers' comments:

Reviewer #1 (Remarks to the Author):

The authors made several revisions that addressed some of my previous questions, but also raise new ones. Overall, there does seem to be a small but significant effect of VIM DBS on sequence learning beyond a general improvement in movement, accompanied by differences in brain oscillations between DBS on/off and repeated vs random sequences. My remaining comments mostly center around presenting the “learning score” more clearly, and the discussion surrounding VIM anatomy.

Major points:

1. The normalized learning score should be defined at least briefly in the Results before describing outcomes.

Based on point 2 below, we have now added the equation defining the normalized learning score to the Methods section. We have also added the following information at the beginning of the Results at line 98:

“For the normalized learning score, a ~~A~~ three-way interaction was observed between *Stimulation Mode* (DBS-ON, DBS-OFF), *Time* (Block 1 to Block 4), and *Tremor Score Difference* ($F(3,24) = 5.84, p = 0.004; \eta^2 = 0.42$), and correcting for *Tremor Score Difference*, the two-way interaction between *Stimulation Mode* and *Time* remained significant ($F(3,24) = 3.70, p = 0.025, \eta^2 = 0.32$; Fig. 1e).”

2. Please clarify the definition of the learning score, preferably with an equation. It is defined “...as the ratio of the difference in mean RT for repeated and random sequences, divided by the mean RT for random sequences”. This is confusing. I think this means $(RT_{\text{repeated}} - RT_{\text{random}}) / RT_{\text{random}}$. However, “the ratio of the difference divided by...” sounds like there are 2 quotients, so I’m not sure what was calculated. If my definition is correct, doesn’t that mean that the DBS on condition was associated with longer RT relative to the random condition across blocks (Fig. 1E)? This needs to be clearly defined since the order in which the difference is taken gives opposite interpretations.

We thank the Reviewer for raising this important point. The equation has been added to the Methods at line 329 to avoid ambiguity and enable clear interpretation of the results:

“To quantify learning, we derived a normalized learning score $\frac{\text{mean RT to random} - \text{mean RT to repeated}}{\text{mean RT to random}}$ ~~defined as the ratio of the difference in mean RT for repeated and random sequences, divided by the mean RT for random sequences.~~ This measure was calculated for each individual for each of the four blocks within a session (DBS-ON or DBS-OFF).”

3. Line 198-200. “Further evidence that lateral VIM is part of a motor network comes from studies showing tactile and kinesthetic responses in lateral VIM but not the medial VIM during movement”. Here, the authors cite a paper describing the lack of kinesthetic/tactile responses in medial VIM, arguing that lateral VIM is part of a motor network while medial VIM is not. However, at least in citation 20, tactile/kinesthetic responses were only tested in the contralateral upper limb. Since there is a face↯arm↯leg somatotopy to VIM moving medial

to lateral, I think it is incorrect to draw the conclusion that lateral VIM is part of a motor network but medial VIM is not. In fact, it seems the strongest evidence that VIM is part of a motor network is the fact that VIM DBS improves tremor.

We thank the Reviewer for pointing out these issues. Our intention was to highlight evidence for a difference between lateral and medial VIM regions, which is consistent with our results, and also to indicate that the kinesthetic/tactile response findings support the lateral VIM being part of a motor network. The absence of this particular evidence for the medial VIM does not, of course, preclude its engagement in motor activity in some other way. We have rewritten this part (see below point 4, as points 3 and 4 are addressed in the same section) and now also include consideration of somatotopy findings in the VIM.

4. Lines 195-196. Similarly, the authors argue that lateral VIM receives more cerebellar afferents than medial VIM. “There are cytoarchitectural and connectivity differences between lateral and medial aspects of VIM, with lateral VIM receiving a large input from cerebellar afferents.” I don’t think this is true. The authors cite two old nonhuman primate papers, but there are many more recent papers in humans using tractography that would be more directly relevant. Generally, VIM thalamus is considered to receive cerebellar afferents, almost by definition. Perhaps part of the problem is that the nomenclature and definitions for thalamic subregions are inconsistent (sometimes based on cytoarchitecture, sometimes based on afferents). For the current paper, the relevant question is whether the medial vs lateral lead locations have different afferents. That seems unlikely since DBS effectively suppressed tremor in all the patients, suggesting that the leads were all in locations with dense cerebellar afferents.

Thank you for drawing our attention to this section. The differing nomenclature, also between the human and animal literature, with “lateral” in the name of the VPLo (VIM in monkeys) adding confusion, as well as not being able to visualize the VIM directly on imaging, make this area complex. Also, by saying the lateral VIM receives cerebellar afferents, we did not mean to exclude medial VIM doing so as well. The Reviewer is right that it could be interpreted that way, though, and we have deleted this phrase. We have only been able to find recent human tractography studies examining connectivity of the VIM as a whole rather than different regions within the VIM, and they also cite earlier animal studies. We have added references to these newer studies. The paragraph starts at line 183:

“The effect on oscillatory power was greater when the stimulation site was at more lateral aspects of the VIM. Moreover, we previously observed enhanced motor sequence learning performance with VIM-DBS delivery to more lateral compared to medial VIM sites⁴. There are cytoarchitectural and connectivity differences between lateral and medial aspects of VIM, with lateral VIM receiving a large input from cerebellar afferents^{17,18}. Further evidence that lateral VIM is part of a motor network comes from studies showing Physiologically, single-cell recordings from the lateral but not medial VIM show tactile and kinesthetic responses in lateral VIM but not the medial VIM during movement^{19,20}, which is relevant for motor learning. Of these recordings, however, 68% of the kinesthetic neurons related to the upper limb and 12% to the more medially represented face location²⁰. The somatotopic organization of the VIM, with the upper limb represented more laterally than the face^{21,22}, could account for their finding of responses only in lateral VIM and our finding that the lateral electrode signals were associated with motor sequence learning tested in the upper limb to a greater

extent than medial electrode signals. ~~learning performance with VIM-DBS delivery to more lateral VIM sites compared to more medial sites⁴.~~

Minor points:

1. Would rephrase “mean normalized learning score” on line 76, or describe it as a measure of learning since at this point in the manuscript the reader won’t know what this means.

We agree with the Reviewer. We have changed this sentence at line 73:

“We examined ~~a behavioral index of motor sequence learning and the mean normalized learning score and~~ scalp electroencephalography (EEG) oscillatory activity during SRTT performance, comparing conditions in which the VIM-DBS was ON or OFF.”

2. I think lines 76-94 are too much detail for the introduction, and moving them later would improve readability. However, I do not feel strongly about this.

In response to the reviews asking about more information in the Introduction, because the Methods section for Communications Biology is at the end of the manuscript, this paragraph is now quite detailed. We have highlighted our responses to the previous round of reviews in purple, and the new changes are in red. While retaining the previously requested information, we have now phrased it more concisely at line 73:

“We examined ~~a behavioral index of motor sequence learning the mean normalized learning score~~ and scalp electroencephalography (EEG) oscillatory activity during SRTT performance, comparing conditions in which the VIM-DBS was ON or OFF. The task required the participants to make a series of finger responses ~~based on the position of a stimulus~~ which either followed a ~~12-item repeated sequence to be learned~~ or ~~were random. was selected in a pseudorandom manner.~~ As our measure of learning, we derived a normalized learning score for each block, ~~comparing contrasting~~ reaction times (RTs) for repeated and random sequences within the block. We then examined oscillatory activity ~~during the SRTT, recorded during the repeated and random sequences,~~ to investigate whether VIM-DBS has an impact on established oscillatory correlates of motor sequence learning. We focused on stimulus-locked activity, using artifact removal methods¹³ that enabled analysis of EEG data recorded during DBS. We used cluster-based permutation tests¹⁴ to examine oscillatory spectral power from 59 electrodes over an epoch starting 200 ms before to 1200 ms after stimulus onset, with the analysis spanning a frequency range of 2-30 Hz. Based on the behavioral results, we examined the general effects of DBS across all trials (repeated and random) at the end of training (Block 4). We then performed separate analyses for repeated and random trials, contrasting the ON-OFF conditions at the end of training. To account for the effects of time, we contrasted trials from Block 4 to Block 1 separately for the repeated and the random sequences, and compared the DBS-ON against DBS-OFF conditions.”

3. Line 169: It’s not clear what “Motor sequence learning was greater...” means. Maybe “faster” or “more efficient”?

Thank you for pointing out this issue, we have now modified this sentence in the Discussion at line 158:

“Motor sequence learning improved ~~was greater~~ when VIM-DBS was on compared to off, with the learning boost during stimulation becoming significant in the final block.”

Reviewer #3 (Remarks to the Author):

The authors have addressed all my concerns. Congratulations for achieving such a difficult study in relatively rare patients.